# Ibrexafungerp, a Novel Triterpenoid Antifungal in Development for the Treatment of Mold Infections

**DOI:** 10.3390/jof8111121

**Published:** 2022-10-25

**Authors:** David A. Angulo, Barbara Alexander, Riina Rautemaa-Richardson, Ana Alastruey-Izquierdo, Martin Hoenigl, Ashraf S. Ibrahim, Mahmoud A. Ghannoum, Thomas R. King, Nkechi E. Azie, Thomas J. Walsh

**Affiliations:** 1SCYNEXIS, Inc., Jersey City, NJ 07302, USA; 2Department of Medicine, Division of Infectious Diseases, Duke University School of Medicine, Durham, NC 27710, USA; 3Mycology Reference Centre Manchester, Wythenshawe Hospital, Manchester University NHS Foundation Trust, Manchester M23 9LT, UK; 4Department of Infectious Diseases, Wythenshawe Hospital, Manchester University NHS Foundation Trust, Manchester M23 9LT, UK; 5Mycology Reference Laboratory, National Center for Microbiology, Instituto de Salud Carlos III, 28029 Madrid, Spain; 6Division of Infectious Diseases, Medical University of Graz, 8036 Graz, Austria; 7David Geffen School of Medicine at UCLA, Los Angeles, CA 90095, USA; 8The Lundquist Institute for Biomedical Innovation at Harbor-UCLA Medical Center, Torrance, CA 90502, USA; 9Department of Pathology, School of Medicine, Case Western Reserve University, Cleveland, OH 44106, USA; 10Center for Innovative Therapeutics and Diagnostics, Richmond, VA 23223, USA

**Keywords:** new antifungal agents, ibrexafungerp, molds, triterpenoid, invasive fungal infection

## Abstract

Molds are ubiquitous in the environment, and immunocompromised patients are at substantial risk of morbidity and mortality due to their underlying disease and the resistance of pathogenic molds to currently recommended antifungal therapies. This combination of weakened-host defense, with limited antifungal treatment options, and the opportunism of environmental molds renders patients at risk and especially vulnerable to invasive mold infections such as *Aspergillus* and members of the Order Mucorales. Currently, available antifungal drugs such as azoles and echinocandins, as well as combinations of the same, offer some degree of efficacy in the prevention and treatment of invasive mold infections, but their use is often limited by drug resistance mechanisms, toxicity, drug-drug interactions, and the relative paucity of oral treatment options. Clearly, there is a need for agents that are of a new class that provides adequate tissue penetration, can be administered orally, and have broad-spectrum efficacy against fungal infections, including those caused by invasive mold organisms. Ibrexafungerp, an orally bioavailable glucan synthase inhibitor, is the first in a new class of triterpenoid antifungals and shares a similar target to the well-established echinocandins. Ibrexafungerp has a very favorable pharmacokinetic profile for the treatment of fungal infections with excellent tissue penetration in organs targeted by molds, such as the lungs, liver, and skin. Ibrexafungerp has demonstrated in vitro activity against *Aspergillus* spp. as well as efficacy in animal models of invasive aspergillosis and mucormycosis. Furthermore, ibrexafungerp is approved for use in the USA for the treatment of women with vulvovaginal candidiasis. Ibrexafungerp is currently being evaluated in clinical trials as monotherapy or in combination with other antifungals for treating invasive fungal infections caused by yeasts and molds. Thus, ibrexafungerp offers promise as a new addition to the clinician’s armamentarium against these difficult-to-treat infections.

## 1. Introduction

Molds are ubiquitous in the environment and pose a serious and growing threat to public health, especially for patients who are immunocompromised [1,2,3]. Invasive fungal infections (IFIs) are associated with mortality rates as high as 30–74% [2], and the associated clinical problem is compounded by the reported increased resistance of these pathogens to azoles, which serve as the first line of pharmacologic treatment against various mold infections. While infections due to opportunistic molds are generally uncommon, the overall incidence is increasing, including infections caused by *Aspergillus* spp. and members of the order Mucorales [2,3]. In addition to invasive pulmonary disease, *Aspergillus* spp. can also cause other syndromes including chronic pulmonary aspergillosis, bronchitis, sinusitis, and hypersensitivity reactions such as allergic bronchopulmonary aspergillosis [4,5]. Moreover, invasive pulmonary aspergillosis (IPA) is a life-threatening infection that occurs more often in immunosuppressed patients, particularly in those with severe and prolonged neutropenia because of myelotoxic chemotherapy, and in those receiving immunosuppressive medication for rejection prophylaxis after organ transplantation or treatment of graft-versus-host disease (GVHD) [6,7,8,9]. IPA has also been associated with severe influenza and COVID-19 infections [10]. IPA has a global prevalence of 42 per 100,000 population, but in the U.S. and Europe, the yearly rate is estimated at 1 to 2 cases per 100,000 [11]. However, this rate is likely to be higher when COVID-19-associated IPA is taken into consideration. Due to the relative severity of IPA, the early initiation of treatment with mold-active azoles is recommended for suspected cases by the Infectious Diseases Society of America (IDSA) guidelines, with liposomal amphotericin B (LAMP) as an alternative [12]. While primary treatment with an echinocandin is not recommended, in some cases, combination treatment with voriconazole may be considered, and micafungin or caspofungin may be used in situations where treatment with an azole is not suitable or contraindicated [12].

Mucormycosis, a rare fungal infection, is associated with mortality rates as high as 50–100% in immunocompromised patients [13,14]. Mucormycosis has a global incidence rate of 0.005–1.7 per million in the population [15], causing infections that are angioinvasive, necrotic, and rapidly progressing. The culprits in mucormycosis are ubiquitous environmental molds of the Mucorales order with global distribution, including the *Rhizopus*, *Mucor*, *Rhizomucor*, and *Lichtheimia* species, as well as the less frequently pathogenic *Syncephalastrum*, *Cunninghamella*, *Apophysomyces*, and *Saksenaea* species [16,17]. Mucormycosis is primarily seen in immunocompromised patients, especially those with uncontrolled diabetes mellitus, prolonged neutropenia, extended use of corticosteroids, and hematological malignancies, patients with injuries such as those caused by road accidents, and more recently has been associated with severe COVID-19 [15,16,18,19]. Mucorales exhibit intrinsic resistance to multiple antifungal agents, and there are limited available treatment options including surgery and amphotericin B-based drugs, and a couple of mold-active azoles [20].

Although still uncommon, increased incidences of fungal infections caused by other rare molds including *Fusarium*, *Lomentospora*, and *Scedosporium* spp., as well as by the dematiaceous molds *Rasamsonia*, *Schizophyllum*, *Scopulariopsis*, *Paecilomyces*, *Penicillium*, *Talaromyces*, and *Purpureocillium* spp., have been reported. The treatment options for these rare mold infections are even more limited because many of them have an inherent resistance to almost all currently available antifungals [21].

Management of mold infections is especially problematic because of the limited number of effective, orally available drugs and increased resistance, especially to triazoles [2,3]. Echinocandins inhibit glucan synthase, the enzyme involved in the synthesis of the fungal cell wall polymer β-(1,3)-D-glucan, and have been previously reported to act against *Aspergillus* spp. and experimental mucormycosis when combined with polyenes [13,14]. The aforementioned IDSA guidelines offer combination antifungal therapy with voriconazole and an echinocandin in certain patients with documented IPA [12]. Marr [22] noted the promise offered by agents that target β-glucan synthase (specifically, echinocandins) in the treatment of aspergillosis as they are not antagonistic, and there is evidence of additive and synergistic activity when coupled with a triazole, presenting the opportunity of adding an echinocandin or other agent with a similar mechanism-of-action target, such as ibrexafungerp, for patients whose aspergillosis has progressed despite triazole monotherapy. Caspofungin was initially approved in 2001 with an indication for the treatment of refractory invasive aspergillosis [23]. Ibrexafungerp, an investigational agent in the treatment of invasive fungal disease, is the first member of the triterpenoid class of antifungals, and acts on the same target as the echinocandins, yet is orally available with evidence of good tissue penetration at many body sites [24].

On the basis of two large, randomized, double-blind, placebo-controlled Phase 3 clinical studies, ibrexafungerp was approved in the United States for the treatment of vulvovaginal candidiasis (VVC) in 2021. In both studies, ibrexafungerp 300 mg BID for 1 day was compared with a placebo. In both the VANISH 303 and VANISH 306 clinical trials, patients receiving ibrexafungerp had significantly higher rates of clinical cures. The cure rates were 50.5% for ibrexafungerp vs. 28.6% for placebo; *p* < 0.001 and 63.3% vs. 44.0%, *p* = 0.007 in VANISH 303 and VANISH 306; respectively, mycological eradication (49.5% for ibrexafungerp vs. 19.4% for placebo, *p* < 0.001 and 58.5% vs. 29.8%, *p* < 0.001 in VANISH 303 and VANISH 306, respectively), and overall therapeutic success (36.0% for ibrexafungerp vs. 12.6% for placebo, *p* < 0.001 and 46.1% vs. 28.4%, *p* = 0.022 in VANISH 303 and VANISH 306, respectively) [25,26]. Most importantly, symptom resolution was sustained, and symptoms improved further at follow-up with ibrexafungerp compared with the placebo. In both studies, ibrexafungerp was generally well tolerated. Adverse effects were primarily gastrointestinal and mild to moderate in severity.

Early evidence of ibrexafungerp in the treatment of invasive mold infections has shown promise, particularly in preclinical models of disease. We review here the potential use of ibrexafungerp for treating invasive infections caused by opportunistic molds.

## 2. Ibrexafungerp Mechanism of Action

Ibrexafungerp (previously MK-3118 and SCY-078; SCYNEXIS, Inc.; Jersey City, NJ, USA) disrupts fungal cell wall synthesis through the inhibition of (1,3)-β-D-glucan synthase [27]. Glucan synthase inhibition is also the mechanism of action of the echinocandins and has been demonstrated to result in a clinically meaningful effect in the treatment of yeast and mold infections. Because ibrexafungerp targets an enzymatic pathway that is not found in human cells, it has a low risk of off-target effects (Figure 1).

Ibrexafungerp, a triterpenoid antifungal, is structurally distinct from the echniocandins (Figure 2) and is a semisynthetic derivative of the naturally occurring hemiacetal triterpene glycoside enfumafungin that incorporates a pyridine triazole at position 15 of the core phenanthropyran carboxylic acid ring system and a 2-amino- 2,3,3-trimethyl-butyl ether at position 14 to improve antifungal potency and pharmacokinetic properties [28]. Compared with echinocandins, ibrexafungerp has the advantages of oral bioavailability, a larger volume of distribution, and its binding to the glucan synthase appears distinct from that of echinocandins, leading to retention of activity against most echinocandin-resistant isolates of *Candida* spp., including those of *C. auris* [29].

## 3. Pharmacokinetics

In animals, ibrexafungerp is absorbed by the gastrointestinal tract after oral administration with a bioavailability of approximately 35–50%. Importantly, ibrexafungerp is widely distributed in tissues and demonstrates excellent tissue penetration in sites commonly associated with invasive fungal infection, including (but not limited to) the lung, liver, kidney, spleen, and skin. Ibrexafungerp has poor penetration into the central nervous system in uninfected animals [24] (Table 1).

Ibrexafungerp is eliminated mainly via metabolism and biliary excretion with <2% of a dose recovered in urine. In vitro studies show that ibrexafungerp is metabolized via hydroxylation by CYP3A4 isoenzymes, followed by glucuronidation and sulfation of a hydroxylated inactive metabolite. Following oral administration of radio-labeled ibrexafungerp to healthy volunteers, a mean of 90% of the radioactive dose (51% as unchanged ibrexafungerp) was recovered in feces, and 1% was recovered in urine. The mean steady-state volume of distribution (V_ss_) of ibrexafungerp in humans is approximately 600 L. Ibrexafungerp is highly protein-bound (greater than 99%), predominantly to albumin [30,31]. A dose-proportional increase in the mean area under the concentration–time curve (AUC_0–∞_) and peak concentration (C_max_) occurred with single doses of 10 mg to 1600 mg within 4 to 6 h, and the mean terminal half-life was 20 to 30 h [27]. The oral bioavailability of ibrexafungerp was increased with a high-fat meal, but absorption was delayed by administration with food [27]. No clinically meaningful effect was observed on the corrected QT interval (QTcF), heart rate, PR, or QRS intervals at plasma ibrexafungerp concentrations up to 4000 ng/mL [32]. Ibrexafungerp concentrations may be increased when administered with strong CYP3A4 inhibitors and dose adjustment may be needed. The coadministration of ibrexafungerp with CYP3A4 inducers is not recommended since it will decrease ibrexafungerp exposure [31].

## 4. In Vitro Activity against *Aspergillus* species

Ibrexafungerp has been extensively tested in vitro against a broad variety of fungal pathogens and has demonstrated potent activity against several isolates, with variable to weak activity against others (Figure 3).

a.Includes *A. fumigatiaffinis, A. thermomutatus, A. udagawae, A. hiratsukae, A. felis, A. citrinoterreus, A. carneus, A. aureoterreus, A. hortai, A. kevei, A. insuetus, A. ochraceus,* and *A. sclerotiorum*.b.No in vitro activity, although in vivo activity has been reported in animal models. 

Glucan synthase inhibitors exhibit a fungistatic effect against many molds [34,35], such as *Aspergillus* spp., despite a high proportion of β-(1,3)-D-glucan in the Aspergillus cell wall [34]. Because glucan synthase inhibitors exhibit a fungistatic effect against *Aspergillus*, MIC values are not accurate; thus, minimum effective concentrations (MEC; the lowest concentration at which abnormal hyphal growth occurs) are used for antifungal susceptibility testing in both CLSI and EUCAST methods [35]. 

In studies of in vitro susceptibility, ibrexafungerp demonstrated potent in vitro activity against *Aspergillus* spp. Complexes including azole-resistant strains [36,37,38,39]. The MEC of ibrexafungerp against a collection of *Aspergillus* spp. (n = 311) ranged from <0.06 µg/mL to 4 µg/mL [36]. The combinations of ibrexafungerp with voriconazole, amphotericin B, or isavuconazole were tested using a checkerboard combination test method against four strains of wild-type (WT) and two strains of azole-resistant *Aspergillus* strains. The combination of ibrexafungerp with voriconazole or isavuconazole showed in vitro synergy against WT *Aspergillus* spp. and was additive against azole-resistant strains. The combination of ibrexafungerp and amphotericin B were synergistic against both WT *Aspergillus* spp. and azole-resistant strains [36]. Similar results were observed in another study reporting that the combination of ibrexafungerp with isavuconazole resulted in a synergistic interaction in *A. fumigatus*, measured by Bliss independence drug interaction analysis [40]. 

The in vitro activity of ibrexafungerp combined with azole antifungals was evaluated against *Aspergillus* spp. isolates from patients undergoing lung transplantation [41]. The median MIC for azoles against all isolates was reduced by ≥4-fold when combined with ibrexafungerp, and the median MIC for ibrexafungerp also was reduced by ≥4-fold when combined with azoles against all isolated other than *A. calidoustus*. A synergistic effect was observed for ibrexafungerp combined with isavuconazole (62%), posaconazole (54%), and voriconazole (53%). Among isolates with an ibrexafungerp or azole MIC < 0.06 µg/mL, combinations remained beneficial with ≥4-fold reduction in MIC for 75%, 50%, and 75% of isolates for isavuconazole, posaconazole, and voriconazole, respectively.

The in vitro activity of ibrexafungerp and other antifungal comparators was evaluated against *Aspergillus* spp. clinical isolates that included both azole-susceptible and azole-resistant *A. fumigatus sensu stricto* (s.s.) and cryptic species with varying degrees of antifungal resistance [42]. The geometric mean MECs against *A. fumigatus s.s.* with ibrexafungerp were 0.040 µg/mL for azole-susceptible strains versus 1.231 µg/mL and 0.660 µg/mL (EUCAST and CLSI) for voriconazole, respectively [43,44]. Against azole-resistant strains, MECs were 0.092 µg/mL and 0.056 µg/mL (EUCAST and CLSI) with ibrexafungerp vs. 2.144 µg/mL and 2.000 µg/mL with voriconazole [43,44]. Ibrexafungerp was also active against most cryptic species of *Aspergillus* [43,44]. Ibrexafungerp exhibited moderate activity against the *A. ustus* species complex, including *A. calidoustus*, *A. insuetus*, and *A. keveii*, but was inactive against *A. alliaceus* (MEC_90_ ≥ 16 µg/mL). Overall, ibrexafungerp demonstrated in vitro activity against many *Aspergillus* spp., including azole-susceptible and azole-resistant strains of *A. fumigatus*.

Additionally, the in vitro activity of ibrexafungerp was examined against a selection of clinical isolates obtained from patients failing azole therapy for chronic pulmonary aspergillosis [45]. The MEC was determined for 22 *A. fumigatus* complex, 3 *A. flavus* complex, and 1 *A. niger* complex isolates with resistance to at least 1 azole antifungal. The MEC range for ibrexafungerp was from 0.008 to 0.25 µg/mL. The ibrexafungerp MEC among isolates considered resistant to all azoles ranged from 0.015 to 0.25 µg/L. One isolate that was resistant to all azoles and amphotericin B was susceptible to ibrexafungerp (MEC 0.125 µg/mL). 

## 5. In Vivo Activity against *Aspergillus* spp. in Animal Models of Infection

In neutropenic mice, ibrexafungerp demonstrated potent in vivo efficacy against wild-type (WT) and azole-resistant strains of *A. fumigatus* when studied in a murine model of invasive aspergillosis [46]. Treatment with oral ibrexafungerp at 7.5 mg/kg/day and 10 mg/kg/day BID significantly increased the mean survival in all strains (*p* ≤ 0.003) and resulted in significant reductions in fungal kidney burden (*p* < 0.05) and serum galactomannan levels (*p* < 0.005). The exposure needed to achieve efficacy was similar to exposures reported in invasive candidiasis models. 

The in vivo efficacy of ibrexafungerp combined with isavuconazole was examined in an experimental neutropenic rabbit model of invasive *A. fumigatus* pneumonia [40]. Ibrexafungerp 2.5 or 7.5 mg/kg/d IV or oral isavuconazole 40 mg/kg/d or their combination were administered. Ibrexafungerp and isavuconazole in combination demonstrated prolonged survival, decreased pulmonary injury, reduced residual fungal burden, and lower GMI and (1,3)-β-D-glucan levels in comparison to those of single therapy for the treatment of invasive pulmonary aspergillosis.

## 6. In Vitro Activity against Other Molds

The in vitro activity of ibrexafungerp and other antifungal drugs was examined in a selection of molds [6,39]. In these studies, ibrexafungerp exhibited activity against *Paecilomyces variotii* (MEC < 0.02 µg/mL to 0.03 µg/mL), *Penicillium citrinum, Neoscytalidium dimidiatum*, *Alternaria* spp., and *Cladosporium* spp. but had limited or no in vitro activity against the Mucorales, *Fusarium* spp., *Purpureocillium lilacinum*, *Acremonium* spp., *Cladosporium cladosporioides, Trichoderma citrinoviride*, and *Trichoderma longibrachiatum* [6,39]. Ibrexafungerp showed variable activity against *Scopulariopsis* spp. and modest activity against *Scedosporium apiospermum* and *Lomentospora prolificans*. Ibrexafungerp was the only drug that exhibited modest in vitro activity against pan-resistant *Lomentospora prolificans* isolates [6].

The in vitro interaction between ibrexafungerp and isavuconazole or amphotericin B was examined in a selection of molds [47]. Ibrexafungerp and isavuconazole exhibited synergistic effects in vitro against *Cunninghamella bertholletiae, S. apiospermum, F. solani,* and *F. oxysporum* with variable results against other molds.

## 7. In Vivo Activity against Other Molds

The in vivo efficacy of ibrexafungerp was examined alone and in combination against the clinical isolates of *Rhizopus delemar* and *M. circinelloides* causing mucormycosis [48]. Immunosuppressed mice were treated with a placebo, ibrexafungerp (30 mg/kg, po, BID), liposomal amphotericin B (LAMB, 10 mg/kg, IV, QD), posaconazole (POSA, 30 mg/kg, po, QD), a combination of ibrexafungerp + LAMB, or a combination of ibrexafungerp + POSA 24 h post-infection, and continued for 7 days for ibrexafungerp and POSA and 4 days for LAMB. The MEC for ibrexafungerp against the *R. delemar* isolate was >8 µg/mL, while the MICs for amphotericin B and posaconazole at 100% inhibition of growth were 0.06 and 0.5 µg/mL, respectively. Survival through Day 21 was the primary endpoint. All mice in the placebo group died by Day 16 post-infection. Following *R. delemar* infection, a significant (*p* < 0.002 for any treatment versus placebo) improvement in the median survival time and overall percent survival was observed with all treatments. In addition, ibrexafungerp + LAMB resulted in significantly (*p* < 0.04) improved overall survival versus any monotherapy. Thus, while ibrexafungerp lacked in vitro activity singly, ibrexafungerp monotherapy demonstrated in vivo efficacy against *R. delemar* infection and was equivalent to other antifungal drugs. The combination of ibrexafungerp + LAMB also demonstrated the highest survival among mice infected with *M. circinelloides*. Tissue burden determined by qPCR was also evaluated after infection with *R. delemar*, and ibrexafungerp mono- or combination-therapy reduced mice lung and brain fungal burden with the combination of ibrexafungerp + LAMB, resulting in the lowest fungal tissue burden when compared with placebo or monotherapy.

## 8. Ongoing Clinical Investigations with Ibrexafungerp for Invasive Mold Infections

There are two ongoing clinical studies evaluating the activity of ibrexafungerp against mold and fungal infections: an open-label study and a blinded phase 2 study. FURI is an open-label study of fungal diseases that are refractory to, or where, patients are intolerant of standard antifungal therapies (clinicaltrials.gov NCT03059992). Eligible patients have proven or probable severe mucocutaneous candidiasis, invasive candidiasis, invasive aspergillosis, or other fungal diseases such as coccidioidomycosis, histoplasmosis, and blastomycosis. In addition, patients must demonstrate treatment failure, intolerance or toxicity related to a currently approved standard-of-care antifungal treatment, or are unable to receive an approved oral antifungal drug (e.g., inadequate susceptibility of the organism), and a continued IV antifungal therapy is clinically undesirable or unfeasible. Most of the patients in this study have yeast infections, but data from some patients with mold infections are expected. Study completion is estimated by the end of 2022.

SCYNERGIA is a phase 2 study evaluating the safety and efficacy of ibrexafungerp co-administered with voriconazole in patients with IPA (ClinicalTrials.gov Identifier: NCT03672292). This is a multicenter, randomized, double-blind, two-arm study in male and female patients 18 years of age and older with a probable or proven IPA. Eligible patients are required to have a probable or proven infection that requires antifungal treatment and a diagnosis consistent with IPA. Patients are randomized to voriconazole alone or voriconazole combined with oral ibrexafungerp. Treatment is continued for a minimum of 6 weeks and up to 13 weeks. Outcomes include safety and tolerability, ibrexafungerp pharmacokinetics, and efficacy evaluated by a composite of clinical, radiological, and mycological response and mortality. Study completion is estimated by the end of 2022.

## 9. Summary

Invasive fungal infections, particularly mold infections, are associated with high mortality and often require prolonged antifungal therapy. Currently, available treatment options for mold infections are scarce and have limitations including increasing reports of resistance development. The increased incidence of *Aspergillus* spp. and Mucorales infections among COVID-19 patients have reminded us of the gaps in the available antifungal treatment armamentarium. Novel antifungals and treatment strategies are needed to improve the outcomes of these difficult-to-treat infections. The novel antifungal ibrexafungerp is a glucan synthase inhibitor with a spectrum of activity that includes several of the molds often involved in systemic diseases. Ibrexafungerp offers oral administration and has been found to be generally safe and well-tolerated. Ibrexafungerp is approved in the United States for treating acute episodic VVC, with approval for recurrent VVC anticipated, and is in development for the treatment of invasive fungal infections. The activity of ibrexafungerp includes the most prevalent fungal pathogens such as *Candida* spp., as well as *Aspergillus* spp. and other molds, and does not have cross-resistance with the azoles, which are the only oral antifungals available for many of these infections. The availability of a once-daily oral dosage form of ibrexafungerp offers the potential to reduce the need for IV administration and may decrease the frequency of prolonged hospitalization and complex dosing schedules, which may enhance the chances of treatment success. The evidence of the activity of ibrexafungerp against mold infections thus far is based on pre-clinical investigations, but these data provide a solid foundation for ongoing clinical development. For *Aspergillus* spp. infections, ibrexafungerp has shown in vitro and in vivo activity as monotherapy and synergetic activity in combination with other antifungals. For Mucorales infections, ibrexafungerp has shown evidence of activity in in vivo models despite the apparent lack of activity in traditional in vitro evaluation. The synergistic activity of ibrexafungerp noted in mice models of mucormycosis when combined with LAMB is particularly interesting considering the high mortality rate for these infections, representing a significant opportunity to improve outcomes for this devastating condition.

When reviewing new therapeutic options under development, it is important to understand their anticipated future role while taking into consideration their attributes and potential clinical implications. Additional clinical data are needed to better define ibrexafungerp’s future role in mold infections; however, the pre-clinical data generated thus far are promising, justifying further development. Combination therapy appears to be an area of opportunity for ibrexafungerp, considering its inherent low risk for drug-to-drug interactions, oral bioavailability, and preclinical data suggesting synergistic effects with other antifungals. The outcomes for many of these mold infections are still far from optimal, and novel therapeutic approaches including antifungal combinations may improve outcomes and warrant further investigation. The activity of ibrexafungerp against azole-resistant *Aspergillus* spp. is another feature of interest, particularly considering patients with chronic *Aspergillus* infections that often require long-term exposure to azoles, eventually developing resistance in geographic areas where azole-resistant *Aspergillus* spp. infections are more prevalent.

Another potential role of ibrexafungerp which will require further clinical demonstration could be its use in preventing fungal infections in patients at heightened risk. Its spectrum of activity, including the most prevalent fungal pathogens such as *Candida* spp. (including azole-resistant and most echinocandin-resistant), *Aspergillus* spp., *Pneumocystis jirovecii*, and Mucorales (based on animal models), together with its oral bioavailability, make ibrexafungerp a good candidate for evaluation as a prophylaxis agent. Based on the available preclinical data, other potential roles of ibrexafungerp include a salvage therapy for patients not responding to, or not tolerating, other antifungal options. As with any antimicrobial, the future role of this novel antifungal should consider the potential for drug resistance development and adequate surveillance and stewardship strategies.

Results from ongoing clinical trials are awaited to further define the efficacy of ibrexafungerp, alone or combined with other antifungals, for treating invasive fungal infections including those caused by *Aspergillus* spp. and other molds.

## Figures and Tables

**Figure 1 jof-08-01121-f001:**
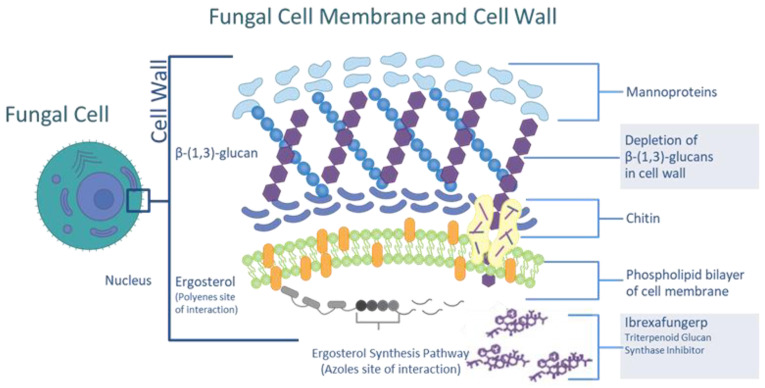
The site and mechanism of action of ibrexafungerp.

**Figure 2 jof-08-01121-f002:**
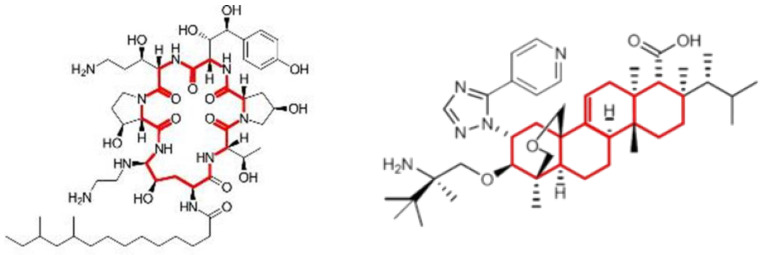
Molecular structures of caspofungin (**left**) and ibrexafungerp (**right**). Red structures indicate core structural components.

**Figure 3 jof-08-01121-f003:**
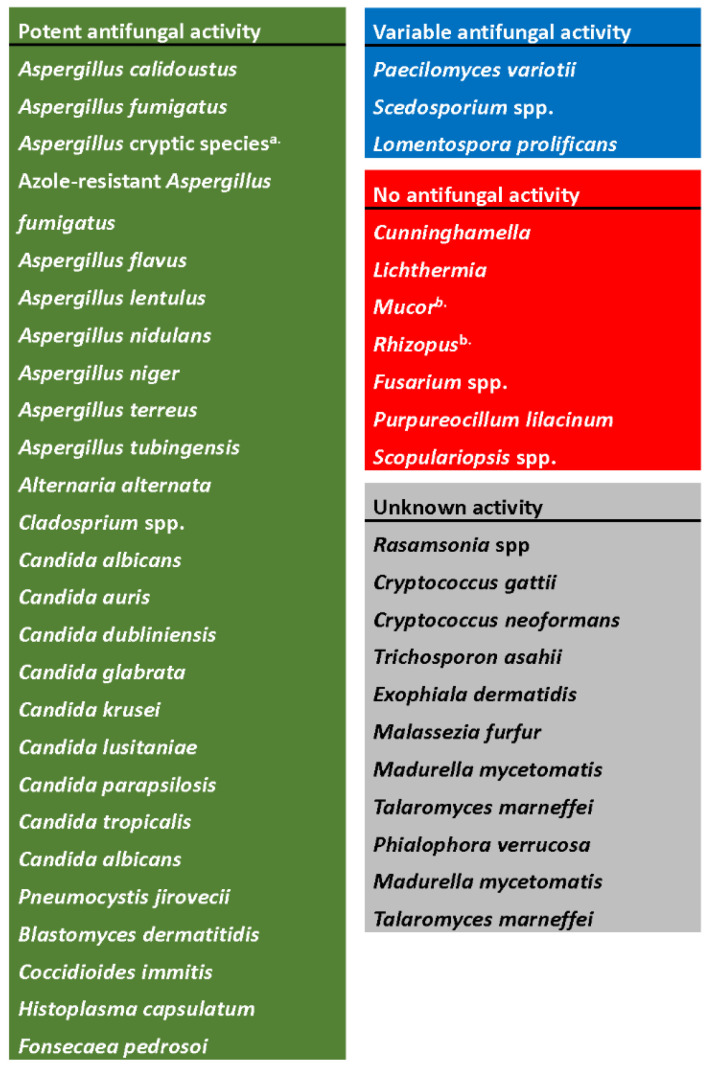
In vitro spectrum of activity of ibrexafungerp. Figure adapted from Hoenigl, 2021 [33].

**Table 1 jof-08-01121-t001:** Mass-balance studies with intravenous ^14^C-ibrexafungerp in albino and pigmented rats.

Tissue	Tissue:Plasma Ratio
Bone	1.3
Bone marrow (femur)	36
Brain (cerebrum)	0.1
Esophagus	6
Eye (uvea)	117
Heart (myocardium)	10
Kidney (cortex)	25
Liver	56.5
Lung	26.5
Lymph node	38
Oral mucosa	5.5
Salivary gland	22.5
Skin (non-pigmented)	11.3
Spleen	75.6
Urinary bladder	7

## Data Availability

Not applicable.

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
