# Peer review of "Ibrexafungerp, a Novel Triterpenoid Antifungal in Development for the Treatment of Mold Infections"

_jof, 2022, doi:10.3390/jof8111121_

Round 1
Reviewer 1 Report
Excellent and thorough review of the latest data that concerns the novel antifungal Ibrexafungerp.
General comments and corrections:
I am curious as why authors have chosen not to include efficacy data for treatment of infection caused by dimorphic molds. Title should maybe be changed for «... Treatment of Opportunistic Mold Infections».
At a few spots in the text I thought it would be relevant to distinguish acquired resistance from that which is inherent/intrinsic (or natural) as it is the case for many pathogenic species such as Fusarium, Scedoporium, Lomentopora and for Mucorales.
Some of the figures were too small in the printed version (fig 1 and 3). The resolution of figure 2 needs to be improved.
Specific comments and corrections
Title: Is IBX «in development» or rather «under evaluation»? Seems to me we're past development phase and more into the clinical efficacy evaluation, unless treatment strategy are being developped. Minor detail.
P2 ln84: «decreased suceptibility» may be too weak to describe the resistance that is often observed. I would suggest «very limited susceptibility or inherently resistant to almost all antifungals».
P3 ln107: Please provide numbers of those higher rates
Figure1: Most of the text in the figure is too small to be readable.
P3 ln132: Add «isolates». Otherwise gives the impression that some Candida spp and C. auris all 100% resistant to echinocandins. I would rephrase «...most Candida spp. echinocandin-resistant isolates, including those of C. auris.»
Figure 2: Resolution could be improved. Why is there a 4 in superscript next to Caspofungin?
Figure 3: Text is too small. «spp» in Scedosporium should not be italicized.
P6 ln173: «most often are used»? MEC is used and recommended by reference methodology of EUCAST and CLSI. Please reword and indicate that both CLSI and EUCAST use MEC.
P6 ln205: A. calidoustus is mispelled.
P7 ln 235: Scytalidium dimidiatum is obsolete. Neoscytalidium dimidiatum is current name (unless it changed again!)
Author Response
REVIEWER 1:
Excellent and thorough review of the latest data that concerns the novel antifungal Ibrexafungerp.
General comments and corrections:
- I am curious as why authors have chosen not to include efficacy data for treatment of infection caused by dimorphic molds. Title should maybe be changed for «... Treatment of Opportunistic Mold Infections».
Thank you for the comments. As the reviewer noted, this paper is focused on the activity of ibrexafungerp against opportunistic molds rather than dimorphic or endemic mycosis. We are aiming to complete ongoing work and summarize the activity against endemic mycosis in a separate publication. We have included the “opportunistic” concept in a couple of places in the introduction to ensure we clarify what the focus of this manuscript is about.
- At a few spots in the text I thought it would be relevant to distinguish acquired resistance from that which is inherent/intrinsic (or natural) as it is the case for many pathogenic species such as Fusarium, Scedosporium, Lomentospora and for Mucorales.
The concept that inherent or acquired resistance has been incorporated in relevant sections of the manuscript.
- Some of the figures were too small in the printed version (fig 1 and 3). The resolution of figure 2 needs to be improved.
We have increased the size of Figures 1 and 3, and replaced Figure 2 with a higher resolution version.
Specific comments and corrections
- Title: Is IBX «in development» or rather «under evaluation»? Seems to me that we're past development phase and more into the clinical efficacy evaluation, unless treatment strategy are being developed. Minor detail.
Response: Thank you for the comment. The authors consider that the term development is still appropriate to refer to the stage of evaluation of the drug. Drug that are not approved for a specific indication can be generally referred as being in “development” stage.
- P2 ln84: «decreased susceptibility» may be too weak to describe the resistance that is often observed. I would suggest «very limited susceptibility or inherently resistant to almost all antifungals».
The Authors agree, and this revision has been made.
- P3 ln107: Please provide numbers of those higher rates
These data have been added.
- Figure1: Most of the text in the figure is too small to be readable.
We have increased the size of Figure 1.
- P3 ln132: Add «isolates». Otherwise gives the impression that some Candida spp and C. auris all 100% resistant to echinocandins. I would rephrase «...most Candida spp. echinocandin-resistant isolates, including those of C. auris.»
We have rephrased this statement accordingly.
- Figure 2: Resolution could be improved. Why is there a 4 in superscript next to Caspofungin?
We have replaced Figure 2 to improve resolution and the superscript has been deleted.
- Figure 3: Text is too small. «spp» in Scedosporium should not be italicized.
We have revised Figure 3 to increase font size, and the text has been corrected.
- P6 ln173: «most often are used»? MEC is used and recommended by reference methodology of EUCAST and CLSI. Please reword and indicate that both CLSI and EUCAST use MEC.
This sentence has been revised accordingly.
- P6 ln205: A. calidoustus is mispelled.
This typographical error has been corrected.
- P7 ln 235: Scytalidium dimidiatum is obsolete. Neoscytalidium dimidiatum is current name (unless it changed again!)
The nomenclature has been revised accordingly.
Reviewer 2 Report
Dear authors,
I read your manuscript concerning the ibrexafungerp and its current applications. The manuscript is an overview of the present research and future clinical applications, well structured and easy to read. I will report some notes to the authors to improve the paper.
1) Line 75, add all the genera in Mucorales.
2) Line 75-78, Injuries and road accidents are one of the fundamental causes of deep mucormycosis.
3) Line 105, 1 day? Do you mean a single dose?
4) Sections 6,7,8 present data that should be summarized in tables.
5) Lines 335-342, You report the perspective to use ibrexafungerp as a preventive therapy in patients subject to an increased risk of fungal infections. Nevertheless, introducing new drugs in clinical practice, especially for preventive purposes, sees the onset of resistance in a short time. This is one of the possible future limitations, which however we have already been seeing for several years in the case of Aspergillus. Moreover, the use of drugs such as echinocandins, which have hepatocolic recycling , can be the basis for selecting resistant strains in the intestine and the one-health approach, evaluating wastewater from hospitals, the onset has been noted related to the use of drugs in the same centres. It would be advisable to introduce these concepts and bring back a one-health perspective linked to ibrexafungerp.
6) Check references style
Author Response
Dear authors,
I read your manuscript concerning the ibrexafungerp and its current applications. The manuscript is an overview of the present research and future clinical applications, well structured and easy to read. I will report some notes to the authors to improve the paper.
- Line 75, add all the genera in Mucorales.
This sentence has been edited to include less-frequently pathogenic species, as well as Rhizomucor.
- Line 75-78, Injuries and road accidents are one of the fundamental causes of deep mucormycosis.
A sentence has been added.
- Line 105, 1 day? Do you mean a single dose?
The dosing for ibrexafungerp in the VANISH studies is correct as written: 300 mg BID (two doses of 300 mg were administered in a single day for a total daily dose of 600mg).
- Sections 6,7,8 present data that should be summarized in tables.
The Authors feel that the information, while exhaustive, is best summarized in text due to its complexity.
- Lines 335-342, You report the perspective to use ibrexafungerp as a preventive therapy in patients subject to an increased risk of fungal infections. Nevertheless, introducing new drugs in clinical practice, especially for preventive purposes, sees the onset of resistance in a short time. This is one of the possible future limitations, which however we have already been seeing for several years in the case of Aspergillus. Moreover, the use of drugs such as echinocandins, which have hepatocolic recycling , can be the basis for selecting resistant strains in the intestine and the one-health approach, evaluating wastewater from hospitals, the onset has been noted related to the use of drugs in the same centres. It would be advisable to introduce these concepts and bring back a one-health perspective linked to ibrexafungerp.
The Authors agree, and following text has been added: “As with any antimicrobial, the future role and use of this novel antifungal should take into consideration a comprehensive approach to attain its optimal usefulness, considering potential for drug resistance development and implementing adequate surveillance and stewardship strategies.”
- Check references style
We have verified that the references are listed in the preferred format of the Journal.
Round 2
Reviewer 2 Report
Dear Authors,
All the corrections have been made.
Author Response
The Authors appreciate the many editorial changes recommended in the second review. We have implemented all, with one exception.
In the second to last paragraph in the paper (lines 371-373), we adapted the suggested wording for better comprehension. It is now written: "As with any antimicrobial, the future role of this novel antifungal should consider potential for drug resistance development and adequate surveillance and stewardship strategies."
We hope that the Journal finds this edit acceptable.